# From Symbolic Perception to Logical Deduction: A Framework for Guiding Language Models in Geometric Reasoning

## Abstract

Plane geometry is a long-standing challenge in AI, requiring the integration of visual perception and mathematical reasoning. Large Multimodal Models (LMMs) such as Gemini 2.5 Pro handles visuo-linguistic inputs but are resource-intensive. We show that a pure Large Language Model, when equipped with specialized modules, can rival state-of-the-art LMMs on complex geometry problems. Our framework integrates a Geometric Vision Parser, which translates diagrams into symbolic form, with a Symbolic Solver that performs formal deductions on angular relations, mitigating hallucinations and promoting interpretable reasoning. To enable rigorous evaluation, we curate a benchmark of difficult problems from the 2025 Chinese Zhongkao examinations, ensuring novelty and testing deeper deductive skills. Experiments demonstrate that our approach achieves performance comparable to Gemini 2.5 Pro while delivering clearer, human-like solutions.

## 1 Introduction

The pursuit of Artificial Intelligence for Mathematics (AI for Math) represents a significant frontier in machine intelligence, with plane geometry standing out as a quintessential grand challenge. Successfully solving geometry problems requires a sophisticated interplay of visual perception and mathematical reasoning—parsing complex diagrams and text, performing logical deductions, and even exhibiting creativity through auxiliary constructions. This process mirrors the progression from perception and cognition to decision-making, marking a critical pathway toward Artificial General Intelligence (AGI). Consequently, geometry problem solving has become a focal point for cutting-edge AI research.

Pioneering works, such as InterGPS (Lu et al., 2021) and AlphaGeometry (Trinh et al., 2024; Chervonyi et al., 2025), demonstrate the power of formal systems in achieving high precision. However, their reliance on formalized languages introduces significant overhead, including the risk of information loss during the translation from naturalistic problems and the inherent brittleness of rule-based systems. Furthermore, their symbolic, non-human-readable outputs pose considerable challenges for user readability. These formal systems are also invariably incomplete; for instance, neither interGPS nor AlphaGeometry can handle inequality or optimization problems in geometry.

The advent of Large Multimodal Models (LMMs), exemplified by systems like Gemini 2.5 Pro, has opened a promising new avenue. By natively processing visuo-linguistic information, LMMs bypass the need for explicit formalization and offer a natural, interactive user experience. Nevertheless, their state-of-the-art performance is not without trade-offs. These models are computationally expensive, resource-intensive, and their reasoning can be opaque. More critically, their performance on specialized domains like geometry is often constrained by distributional shifts between their generalist training data and the specific symbolic logic of geometric diagrams.

This paper challenges the prevailing assumption that end-to-end LMMs are the definitive solution for complex geometric reasoning. We posit that a powerful Large Language Model (LLM), when augmented with specialized geometric perception and deductive reasoning capabilities, can not only match but surpass the performance of leading LMMs. While LLMs like Deepseek-R1 possess immense abstract reasoning power, their inherent lack of visual processing has rendered them unsuitable for such tasks. Our work directly addresses this limitation. We introduce a novel framework

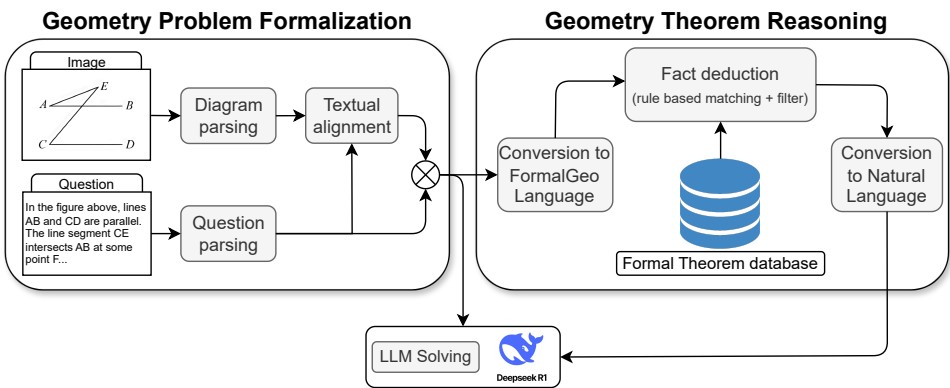

Figure 1: Schematic representation of the pipeline.

designed to empower LLMs for advanced geometric problem-solving. At its core are two critical components:

A Geometric Vision Parser, which translates unstructured diagram images into a symbolic, structured representation. By integrating OCR and geometric primitive detection, it provides the LLM with the precise, high-fidelity visual information it naturally lacks.

A Symbolic Solver module, which performs targeted formal deduction on angular relationships—a frequent source of LLM hallucination. This not only enhances the model's perceptual accuracy but also critically guides it towards more elegant, deductive solutions, steering it away from brittle, brute-force coordinate calculations that plague many current models and degrade readability.

Figure 1 illustrates the overall workflow. The system first converts problem statements and associated diagrams into a formal representation of geometric entities and constraints. Next, the reasoning engine predicts relevant theorems by combining search-based matching and rule filtering, producing a sequence of theorem applications. These intermediate results are translated into natural language and provided to the LLM, which synthesizes them into a complete solution with both deductive rigor and explanatory clarity.

To rigorously evaluate our approach against the state-of-the-art, we identified a critical gap in existing benchmarks. Datasets like GeoQA and Geometry3K, while foundational, are often saturated in performance and may suffer from data contamination, having been publicly available for years. Appendix A.2 provides a few suspected cases of data contamination in Gemini-2.5-Pro (Table 5).

More importantly, their structural simplicity fails to challenge models with complex multi-step deductions or the necessity for auxiliary line constructions. To address this, we introduce a new, challenging benchmark derived from 2025 Chinese Zhongkao (national middle school examination) math problems—a source of difficult problems that demand deep reasoning and creative insight, while also guaranteeing data novelty. Since these questions are from recent public examinations, they are highly unlikely to have been included in the pre-training data of existing large models, thus enabling a fair evaluation of geometric reasoning capabilities without the risk of data contamination.

Our contributions are threefold:

- We propose a novel framework that, for the first time, enables a LLM (R1) to achieves comparable performance to a state-of-the-art LMM (Gemini 2.5 Pro) on complex, unseen plane geometry problems.

- We introduce a new, high-difficulty benchmark curated from 2025 Chinese Zhongkao examination questions, providing a more challenging and reliable testbed for future research in geometric reasoning.

- New Geometric Vision Parser and Symbolic Solver modules that guide LLMs towards human-like, interpretable reasoning by substantially reducing the model's reliance on non-

intuitive, "brute-force" coordinate geometry, thereby enhancing the readability and user-friendliness of the solutions.

# 2 RELATED WORK

Solving plane geometry problems with AI has been a long-standing challenge, requiring both sophisticated multimodal understanding of text and diagrams, and rigorous logical deduction. Early efforts relied on symbolic systems, which have gradually given way to data-driven neural networks and, more recently, powerful hybrid architectures that integrate the strengths of both paradigms.

## 2.1 SYMBOLIC-BASED METHODS

Early approaches to automated geometry problem solving were dominated by symbolic and rule-based systems. These methods prioritize logical soundness and interpretability. They typically employ meticulously crafted rule-based parsers to convert the natural language text into a formal, symbolic language. The diagrammatic primitives are also converted into formal descriptions through manual translation or a trained diagram parser. PGDP-Net (Zhang et al., 2022) models geometric element recognition as an entity segmentation task and uses a Graph Neural Network (GNN) to identify element classes and their relationships, outputting a formal language description. The problem is then solved using symbolic reasoners that operate within this formal system. Pioneering works like Inter-GPS and Formal-Geo (Zhang et al., 2024) are exemplars of this approach. While powerful in logical consistency, their reliance on hand-crafted rules can limit their scalability.

## 2.2 NEURAL NETWORK-BASED METHODS

With the rise of deep learning, researchers began exploring end-to-end neural models that learn to solve geometry problems directly from data. These methods encode the problem into dense vector representations, utilizing neural networks to predict solutions based on the learned features. For instance, NGS (Chen et al., 2021) employed a text encoder to learn textual features of the problem, and introduced auxiliary tasks like a jigsaw puzzle game to learn robust visual representations of geometric diagrams. Building on this, models like PGPSNet (Zhang et al., 2023) utilized dual-stream encoders with bidirectional GRUs to fuse textual and visual features. LANS (Li et al., 2023) further refined this architecture by introducing a layout-aware pre-training module and employing contrastive learning to enhance the alignment between visual points and textual tokens, leading to more precise multimodal understanding before feeding the fused representation to a sequence-to-sequence decoder.

## 2.3 HYBRID METHODS

To combine the logical rigor of symbolic systems with the rapid decision-making strengths of neural networks, a significant body of work has focused on hybrid, or neuro-symbolic, methods. A dominant paradigm in this area involves using neural networks as theorem prediction modules to accelerate inference—addressing the fact that traditional symbolic approaches often rely on time- and resource-intensive search processes. For example, GeoDRL (Peng et al., 2023) represents the problem as a geometric logic graph and trains a Graph Neural Network (GNN) via reinforcement learning to act as a policy network, which selects the most promising theorem to apply within a formal reasoning system at each step of the reasoning process.

## 2.4 RECENT TRENDS WITH LARGE MODELS

Recently, LLMs have been adopted as powerful backend reasoners, with methods like GeoX (Xia et al., 2024), DFE-GPS (Zhang et al., 2025), and Pi-GPS (Zhao et al., 2025) leveraging them for the final solving stage. The focus has now shifted towards augmenting these large foundational models with specialized strategies to enhance their geometric reasoning capabilities. These strategies include refining the formal problem representation (Ping et al., 2025), generating large-scale annotated datasets (Wu et al., 2025), improving inference-time reasoning with advanced search algorithms (Wang et al., 2025b), and using reinforcement learning to tackle notoriously difficult sub-problems like auxiliary line construction (Wang et al., 2025c).

# 3 METHOD

Solving geometry problems, especially those that heavily rely on diagrams, requires a precise understanding of visual geometric relationships. Unlike algebraic or purely text-based tasks, geometric problem solving often depends on recognizing implicit constraints, such as collinearity, angle properties, that are visually encoded in diagrams rather than explicitly stated in text. However, current multimodal models exhibit significant limitations in accurately parsing and reasoning about geometric diagrams (Wang et al., 2025a), which poses a major challenge for automated geometry problem solving. We propose a hybrid framework for automated geometry problem solving that integrates formalized problem parsing, theorem-driven reasoning, and LLM inference. Unlike traditional symbolic geometry solvers, our approach leverages both rule-based theorem application and the generative capabilities of LLMs to produce coherent, human-readable solutions. As shown in Figure 1, the framework consists of three core modules: (1) Geometric Problem Formalization, (2) Geometric Formal Reasoning, and (3) LLM-based Answer Generation.

## 3.1 GEOMETRIC PROBLEM FORMALIZATION

Solving geometry problems requires a precise formal representation of both diagrammatic and textual information. Inspired by Inter-GPS (Lu et al., 2021), we first design a bottom-up diagram parsing pipeline for extracting and constructing geometric information from diagrams. A visual representation of this pipeline is shown in Figure 2.

The pipeline begins with the extraction of geometric primitives, which serve as the foundational elements of the diagram representation. A primitive is defined as basic geometric elements such as a point, line, circle, or arc segment. We start by using a fine-tuned YOLO pose model to identify geometric line segments (Khanam & Hussain, 2024). This is followed by a "primitive finder" step that merges overlapping or closely aligned detections into consistent line entities. During this phase, we also detect more primitives by using Hough transforms to find circles and angular structures. Points and angles are then located based on the intersections between the detected lines and circles.

Beyond geometric primitives, the parser also identifies diagrammatic symbols (such as angle marks and right-angle indicators) and textual annotations (such as point labels or length values). Symbol and text regions are first localized using a finetuned YOLO detection model. The content in the text regions is then recognized with a specialized recognition model (Texteller[1]). At this stage, we obtain a primitive set $P = \{p_1, p_2, \ldots, p_m\}$ and a symbol set $S = \{s_1, s_2, \ldots, s_n\}$. To create meaningful connections between these components, we assign each symbol with its corresponding primitive using a greedy assignment strategy. This process is formally defined as:

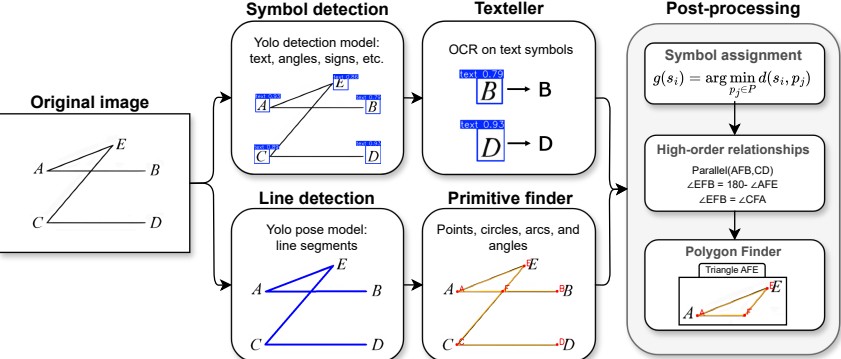

Figure 2: Diagram parser pipeline. From this pipeline we extract relevant primitives and relationships from the diagram image.

$$g(s_i) = \arg\min_{p_j \in P} d(s_i, p_j), \quad s.t. \ (s_i, p_j) \in \text{Feasibility set } F.$$

---

[1] https://github.com/OleehyO/TexTeller

Here, $d(s_i, p_j)$ represents the Euclidean distance between the detected location of symbol $s_i$ and the candidate primitive $p_j$. A key constraint is that the symbol $s_i$ and primitive $p_j$ must be compatible, belonging to a defined feasibility set $F$. For example, the text symbol "10°" can only be assigned to angle or arc measure primitives, and the perpendicular symbol can only be assigned to orthogonal line primitives. This ensures that each symbol is assigned to the closest compatible primitive.

We further introduce post-processing steps to derive higher-order geometric relations and align them with textual descriptions. Such enriched representations are essential for guiding the LLM in subsequent reasoning stages. While the primitive-level elements (e.g., points, lines, circles) provide a structural foundation, additional relational information enables a more expressive and semantically meaningful formalization. For example, besides extracting simple relationships like `PointLiesOnLine(Point(F),Line(A,B))`, which only indicates a collinearity relation, we also encode collinear relationships in a systematic left-to-right, top-to-bottom order to fully capture the spatial relationships between points. In addition, we extend the relationship list to incorporate cyclic relations, polygonal structures, and angle dependencies. A detailed catalog of these formal commands, along with additional examples, is provided in the Appendix A.2, Table 4.

Diagrams in geometry problems are often just schematic sketches, not precise drawings, so their visual information can be redundant or even misleading. To ensure we only use relevant and correct geometry information, after diagram parsing, we perform a step called "textual alignment" (see Figure 1) to align the visual data with the text. We utilize a rule-based parsing strategy to filter and refine the information extracted from the diagram image. For example, if the problem text states that the figure contains a rectangle $ABCD$, our system automatically adds the defining properties of a rectangle —such as opposite sides being parallel ($AB \parallel DC$) and adjacent sides being perpendicular ($AD \perp AB$) —to its list of known relationships. This approach guarantees that our system's reasoning is based on correct geometric properties derived from the text, not on potentially inaccurate visual relationships extracted from the image.

## 3.2 GEOMETRIC FORMAL REASONING

While LLMs have demonstrated remarkable capabilities in complex reasoning, they frequently produce hallucinations, particularly in problems concerning angles. The core of this issue lies in the LLM's reliance on analytical methods, such as constructing parametric equations, to validate steps. This paradigm fails to capture fundamental, non-analytic geometric knowledge. For example, analytic geometry can confirm the equality of vertical angles ($\angle 1 = \angle 2$) via dot products but cannot intrinsically represent the concept of "being vertical angles". This relationship is defined axiomatically ("opposite angles at an intersection"), a form of knowledge that LLMs struggle to ground.

To bridge this gap, we propose a Formal Reasoning Module that deduces angle properties based on formal geometric rules. This module acts as an external verifier, providing the LLM with reliable geometric facts for its subsequent reasoning. The process includes three key steps: Literal Expansion, Fact Deduction, and Fact Filtering.

We first process the diagram parser's output. This involves not only translating the elements into the FormalGeo symbolic language but also augmenting the representation by inferring implicit relationships. For instance, we expand collinear relations (e.g., `Collinear(A,B,C) & Collinear(B,C,D)` implies `Collinear(A,B,C,D)`) and composite angles (e.g., `Collinear(A,B,C) & Angle(D,A,C)` implies `Angle(D,A,B)`). This augmentation provides a richer set of primitives and relations for the next step, reducing inferential gaps.

Secondly, we match the parsed geometric elements against the input parameters of rules defined in our theorem library to derive new facts. However, an exhaustive matching approach, which iterates through every possible combination, is computationally prohibitive. As illustrated in Appendix A.2, Figure 6, a simple diagram with 7 points may yield 21 distinct line segments. To apply a theorem that requires four specific line segments, such as an "M-theorem" defined as `geometric_m_model_angle_equation(AB,BC,CD,DE)`, a brute-force approach would need to evaluate at least $21 \times 20 \times 19 \times 18 = 143,640$ permutations, resulting in an intractably large search space. To mitigate this combinatorial explosion, we introduce constraints by defining strict ordering rules for points within our theorem definitions. We further leverage geometric priors, such as parallel and perpendicular relationships, to prune invalid argument combinations. Through these heuristics, we successfully reduced the number of candidate permutations.

In the final step, we assess the relevance of the derived conclusions to the original problem statement. Only the pertinent conclusions are selected and provided as supplementary input to the LLM.

### 3.3 LLM-BASED ANSWER GENERATION

The final stage of our framework leverages a LLM to produce coherent and human-readable solutions. To achieve this, we synthesize the outputs from all preceding modules to construct a comprehensive, task-specific prompt. This prompt is meticulously structured to integrate several key streams of information: (1) the verbatim textual description of the geometry problem, (2) the parsed spatial and logical relationships between geometric entities and the pixel coordinates of key points, which serve as a reference for visual grounding, and (3) the set of high-relevance geometric deductions produced and filtered by our Formal Reasoning Module. By assembling the input in this manner, we create an augmented prompt that grounds the LLM in the specific context of the problem. More critically, it provides the model with verified conclusions, effectively constraining its reasoning space and mitigating the risk of hallucination. This enriched input is then fed to the LLM to generate the final, step-by-step derivation.

## 4 ZHONGKAO GEOMETRY BENCHMARK

As the reasoning capabilities of LLMs and LMMs continue to advance, the need for robust and challenging evaluation benchmarks becomes increasingly critical. We posit that a high-quality benchmark must satisfy three core principles: Quality (well-posed problems with clear descriptions), Difficulty (requiring a significant number of reasoning steps or solving for multiple objectives), and Diversity (broad coverage of problem types and concepts).

However, existing plane geometry benchmarks are becoming saturated, failing to adequately challenge the capabilities of state-of-the-art models. For instance, the GeoQA (Chen et al., 2021) training set is heavily skewed towards angle and length calculations, with only 267 of its 3,509 problems involving area or other concepts. Its complexity is limited, with an average of just 1.96 reasoning steps (max 4). While the GeoQA+ (Cao & Xiao, 2022) dataset improves on diversity, it remains constrained in difficulty, with an average of 2.23 reasoning steps (max 8). [2]

To address these limitations, we introduce a new benchmark suite, systematically sourced from recent Chinese Zhongkao Examination, which are renowned for their quality, complexity, and diversity. Furthermore, we contend that a fourth crucial principle for modern benchmarks is Contamination Prevention. The risk that a model has been exposed to test data during its pre-training phase poses a significant threat to the validity of the evaluation.

Adhering to these four principles, we designed a three-tiered evaluation suite for plane geometry with progressively increasing difficulty and reduced risk of data contamination:

**ZhongkaoGeo-L1.** Sourced primarily from official and mock examination papers from 2023 and earlier. This dataset covers a wide spectrum of formats (e.g., multiple-choice, calculation, and proof problems) and assesses diverse concepts, including: angle calculation, length (ratio/perimeter/trigonometric-value) calculation, area (ratio) calculation, comprehensive problems (requiring a combination of the above), composite transformations (rotation, translation, symmetry, dynamic points), and optimization (finding extremal values).

**ZhongkaoGeo-L2.** Sourced from examinations administered between 2024 and early 2025. This dataset increases complexity by featuring a higher proportion of problems with multiple sub-questions and a greater emphasis on difficult problem archetypes (37% vs. 30% on Knowledge type D+F+G). See Table 1 for a detailed comparison.

**ZhongkaoGeo-L3.** Comprises plane geometry problems from the official 2025 examinations in Chinese provincial capitals and major municipalities. This dataset consists of entirely novel data, providing the most reliable assessment of a model's un-memorized reasoning capabilities.

After a meticulous curation process, where we discarded problems with unclear images, text-diagram mismatches, or missing/ambiguous solutions, our final benchmark suite consists of 89

---

[2]We report statistics from the training sets as presented in the original papers; their random-split methodology ensures these distributions are representative of the test sets.

Table 1: A detailed statistical comparison of the ZhongkaoGeo-L1&L2 datasets, highlighting the increased difficulty of our proposed benchmark. In each data block, the top row shows the distribution of problems by the number of sub-questions ('1' to '5' and their average), while the bottom row shows the distribution by knowledge type ('A' to 'F'). The knowledge types are defined as A: Angle, B: Length, C: Area, D: Comprehensive, E: Transformation, and F: Optimization.

| Dataset | # Problems | Block | Counts | | | | | |
|---------|-----------|-------|--------|---|---|---|---|---|
| | | | 1/A | 2/B | 3/C | 4/D | 5/E | Avg./F |
| ZhongkaoGeo-L1 | 89 | Sub-questions | 69 | 15 | 5 | 0 | 0 | 1.281 |
| | | Knowledge types | 17 | 38 | 7 | 7 | 12 | 8 |
| ZhongkaoGeo-L2 | 83 | Sub-questions | 46 | 14 | 17 | 5 | 1 | 1.807 |
| | | Knowledge types | 28 | 20 | 4 | 9 | 17 | 5 |

problems in ZhongkaoGeo-L1, 83 in ZhongkaoGeo-L2, and 105 in ZhongkaoGeo-L3. Notably, unlike traditional plane geometry benchmarks which typically present a single problem with one question and one solution target, the problems in our benchmark are often structured as multi-step tasks, containing several related sub-questions that require achieving multiple solution targets.

## 5 EXPERIMENTS

In this section, we conduct a series of experiments to evaluate the effectiveness of our proposed method on three datasets of increasing difficulty: ZhongkaoGeo-L1, ZhongkaoGeo-L2, and ZhongkaoGeo-L3. Our method is built on Deepseek-R1.

### 5.1 BASELINES

To rigorously evaluate our approach, we compare it against a suite of state-of-the-art Large Models, renowned for their powerful reasoning and general problem-solving capabilities. Our selection prioritizes models that have demonstrated leading performance on various general-purpose benchmarks, ensuring a high-standard and relevant comparison. The chosen baselines are: GPT-o1, Qwen3-32B & Qwen3-235B, DeepSeek-R1, and Gemini 2.5-Pro, the most recent high-performance model possessing strong multi-modal and complex reasoning capabilities.

### 5.2 EVALUATION METRICS

For the L1 and L2 datasets, we use a strict accuracy metric. A question is marked as correct (score of 1) only if the generated answer perfectly matches the ground truth. Any deviation, however minor, results in a score of 0. The final accuracy is calculated as the ratio of correctly answered questions to the total number of questions.

Since the L3 dataset is directly sourced from the Zhongkao examinations held this year, we adopt a Scoring Rate metric based on the official grading rubrics of the Zhongkao examinations. Each sub-question within a problem is assigned a specific point value. A model receives the full points for a sub-question if its answer is correct and zero points otherwise. The total score for a problem is the sum of scores from its constituent sub-questions. The final Scoring Rate is the ratio of the total score achieved by the model to the maximum possible score across the entire dataset.

### 5.3 IMPLEMENTATION DETAILS

All experiments are conducted using the official APIs or publicly available weights of the baseline models to ensure fair comparison. The input for LLMs was solely the textual description of the problem, while for LMMs, it included both the image and the text. For the closed-source models, a temperature of 0.1 was employed. For the open-source models, we followed the official recommendations and used the default parameters, reporting the average performance over three runs. To assess performance stability and potential gains from ensembling, we also report results for "Ours (Major@3)". This strategy involves performing three independent inference runs and selecting the

most frequent answer as the final output through a majority vote. The prompt for our full pipeline is designed as `Prompts with Geometric Information` in Appendix A.3, while for all other situations, we use the conventional `Problem-Solving Prompts`.

## 5.4 RESULTS AND ANALYSIS

Table 2: Performance (Accuracy) on ZhongkaoGeo-L1&L2

| MODEL | ZhongKaoGeo-L1 | ZhongKaoGeo-L2 |
|---|---|---|
| GPT-o1 | 82.02 | 56.63 |
| Qwen3-32B | 84.64 | 64.26 |
| Qwen3-235B | 86.52 | 69.48 |
| DeepSeek-R1 | 88.39 | 67.87 |
| Gemini2.5-Pro | **94.38** | 73.49 |
| Ours | 92.13 | 74.30 |
| Ours (Major@3) | 93.26 | **78.31** |

**Performance on ZhongkaoGeo-L1 & L2.**   The comparative results on the L1 and L2 datasets are presented in Table 2. Our method demonstrates highly competitive performance. In its standard configuration ("Ours"), it achieves an accuracy of 92.13% on L1 and 74.30% on L2, outperforming most SOTA baselines, including the large-scale Qwen3-235B and DeepSeek-R1 models. Crucially, when employing the majority voting strategy ("Ours (Major@3)"), our model sets a new state-of-the-art on both datasets, reaching 93.26% on L1 and a remarkable 78.31% on L2. This result not only surpasses the strongest baseline, Gemini 2.5-Pro, but also highlights the robustness and reliability of our approach. The significant performance drop from L1 to L2 across all models underscores the increased difficulty and complexity of the L2 dataset. This further reveals the higher risk of data leakage associated with earlier benchmarks. Notably, our method exhibits strong generalization, with Major@3 widening the performance gap over competitors on the more challenging L2 set.

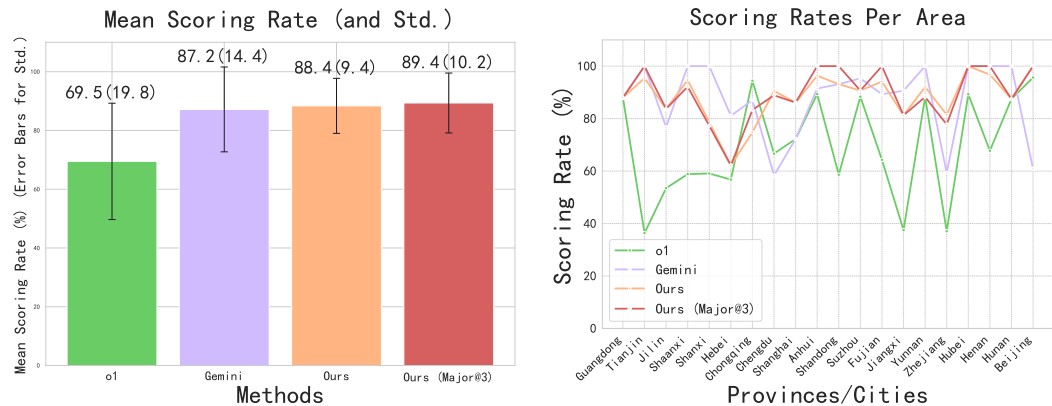

Figure 3: Performance (Scoring Rate) on ZhongkaoGeo-L3. We also report the standard deviation of score rates computed over different regional exam papers.

**Performance on ZhongkaoGeo-L3.**   Figure 3 presents the results on the most challenging ZhongkaoGeo-L3 dataset. The bar chart on the left illustrates the mean scoring rate. Our method ("Ours") achieves a mean scoring rate of 88.4%, which is already superior to Gemini 2.5-Pro's 87.2%. With majority voting, "Ours (Major@3)" further improves the score to 89.4%, establishing a clear advantage over all baselines. Furthermore, the error bars, representing standard deviation,

indicate that our method's performance is stable, with a variance comparable to that of Gemini 2.5-Pro. The line plot on the right provides a more granular, per-province/city view of the scoring rates. This plot reveals that while all models exhibit performance fluctuations depending on the specific exam questions, our method (orange and red lines) consistently tracks or exceeds the performance of the best baseline (Gemini 2.5-Pro, purple line). This fine-grained analysis confirms that our model's superiority is not due to overfitting on specific question types but rather a consistent, high-level reasoning capability across a diverse range of problems.

In summary, the experimental results across all three datasets consistently demonstrate that our proposed method, particularly with the Major@3 enhancement, achieves state-of-the-art performance, outperforming even the most advanced proprietary large multimodal models.

## 5.5 ABLATION STUDY

### 5.5.1 ANALYSIS OF GEOMETRIC PARSING PERFORMANCE

Table 3: Performance comparison on the parsing of implicit geometric relations on the ZhongkaoGeo-L3 dataset. We report the number of problems where each type of relation was correctly identified. "All Relations Correct" is a holistic metric counting problems where all three relation types were successfully parsed. Our method demonstrates a substantial improvement across all categories, highlighting its superior diagram understanding capabilities.

| Relation Type | Qwen-VL-2.5-72B | Ours | Improvement ($\Delta$) |
|---|---|---|---|
| Collinearity | 45 | **85** | +40 |
| Concyclicity | 74 | **95** | +21 |
| Angular Position Relations | 46 | **86** | +40 |
| **All Relations Correct** | 23 | **81** | **+58** |

To solve geometry problems, a model must parse implicit visual relations from diagrams. We evaluated our method's ability to do this on the ZhongkaoGeo-L3 dataset, comparing it against the strong Qwen-VL-2.5-72B baseline. The results, shown in Table 3, unequivocally demonstrate our method's superiority. Component-wise, our model nearly doubles the baseline's performance in identifying individual relations like collinearity and angular positions. More critically, for holistic understanding (parsing all relations in a problem correctly), our model achieves a score of 81 compared to the baseline's 23—a greater than 3.5-fold improvement. This substantial leap shows that our method consistently constructs a comprehensive and accurate symbolic representation of the geometric figure. While baseline models often miss necessary conditions, our approach provides a much stronger foundation for successful, multi-step reasoning.

### 5.5.2 EFFECT OF GEOMETRIC FORMAL REASONING

With the help of Geometric Formal Reasoning, our method is available to guide the model to a readable solution instead of the brute-force coordinate method, as shown in Appendix A.2, Figure 4 5.

## 6 CONCLUSION

This work challenges the prevailing paradigm that ever-larger multimodal models are the sole path to advancing geometric reasoning. We have demonstrated that a pure Large Language Model, when augmented with a specialized Geometric Vision Parser and a Symbolic Solver, can achieve and even surpass the performance of a state-of-the-art LMM like Gemini 2.5 Pro. Our introduction of the challenging, contamination-free ZhongkaoGeo benchmark provides a rigorous new standard for evaluating true deductive capabilities. More importantly, by promoting human-like deductive reasoning over opaque, brute-force methods, our framework advances the development of transparent and interpretable AI for mathematical problem-solving.

## ETHICS STATEMENT

This work does not raise any ethical concerns.

## REPRODUCIBILITY STATEMENT

To ensure the reproducibility of this work, we will make all necessary materials openly available upon paper acceptance or by earlier request from the reviewers. This will include the ZhongKaoGeo Benchmarks and the code for the pipeline and all experiments in this paper.

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

# A APPENDIX

## A.1 LLM USAGE STATEMENT

In this work, the primary application of a Large Language Model (LLM) is as the backend reasoner in the final Answer Generation stage of our proposed framework. The LLM's role is to synthesize the structured output from the Geometric Vision Parser and the formal deductions from the Symbolic Solver into a coherent, human-readable solution. Additionally, several state-of-the-art LLMs and LMMs were benchmarked to provide a comprehensive comparative analysis. While the LLM is an integral computational component and was used for grammatical refinement of the manuscript, all research ideation, framework design, and analysis of results were conducted exclusively by the human authors.

## A.2 TABLES & FIGURES

Table 4: Examples of higher-order predicates extending the basic FormalGeo representation.

| Predicate | Description and Example |
|---|---|
| `PointsLieOnCircle(P_1, P_2, ..., P_k, Circle(O, r))` | All listed points lie on the same circle, given in clockwise order. Example: `PointsLieOnCircle(C, A, B, D, Circle(O, radius_0))` |
| `Collinear(P_1, P_2, ..., P_k)` | All listed points lie on the same straight line, arranged in left-to-right and top-to-bottom order according to the diagram. Example: `Collinear(A, O, B, F)` |
| `Shape(P_1, P_2, ..., P_k)` | Defines a polygonal figure (triangle, quadrilateral, etc.), with vertices specified in clockwise order. Example: `Shape(A, F, D)` |
| `Angle(P_i, P_j, P_k) = Angle(...)` | Encodes equalities or additive relations between angles. Example: `Angle(B,A,C) = Angle(B,A,D) + Angle(C,A,D)` |

Table 5: Suspected cases of data contamination on the Mathvista test dataset. Gemini-2.5-Pro corrects erroneous Diagram Parsing (DP) information, appearing to infer details from the original image despite not being provided with it. The model's corrections are highlighted in red.

| Image | Question | DP Information | Gemini-2.5-Pro Answer |
|---|---|---|---|
|  | As shown in the figure, triangle ABC is the inscribed triangle of circle O, then the degree of angle ACB is... | ..."Formulas": ["MeasureOf( Angle(B,O,A), 35° )", "Angle(B,A,C) = Angle(B,A,O) + Angle(C,A,O)"... | From the figure, we are given that the measure of angle OAB is 35°... |
|  | As shown in the figure, chord AC is parallel to radius OB, then angle OAB is... | ..."Formulas": ["MeasureOf(Angle(O,B,P), 50% )", "Angle(C,A,O) = Angle(O,A,P) + An... | From the figure, the central angle ∠BOC is given as 50°... |
|  | As shown in the figure, AB is parallel to CD, then the degree of angle 1 is... | [No information about angles given or 135°] | The figure shows an angle of 135°... |

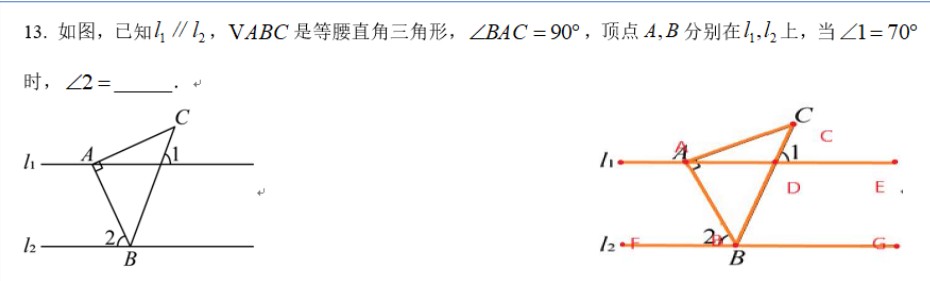

**Geometry Problem:** As shown in the figure, given $l_1 \parallel l_2$, $\triangle ABC$ is an isosceles right-angled triangle with $\angle BAC = 90°$. Vertices $A$ and $B$ are on lines $l_1$ and $l_2$ respectively. When $\angle 1 = 70°$, find the measure of $\angle 2$.

**Image Parsing Result:** ...   (Correct structured data) ...

*[... 22k+ tokens of reasoning omitted ...]*

1. **Coordinate System Setup:** Let $l_1$ be $y = 0$ and $l_2$ be $y = -h$. Let $A = (0,0)$ and $B = (b, -h)$. Then $C = (h, b)$.

2. **Equation of Line BC:** The slope of line BC is $\frac{b+h}{h-b}$. The intersection $D$ of line BC with $l_1$ is $(\frac{h^2+b^2}{h+b}, 0)$.

3. **Analysis of $\angle 1 = 70°$:** The cosine of the angle between vector $\overrightarrow{DC}$ and the positive x-axis is calculated, yielding $\cos 70° = \frac{h-b}{\sqrt{2(h^2+b^2)}}$.

4. **Parameter Solving:** Let $t = h/b$. Solving the equation gives $t \approx 2.146$, so $h \approx 2.146b$.

5. **Calculation of $\angle 2$:** $\angle 2$ is the angle between AB and the horizontal line $l_2$. Its tangent is $\tan\theta = \frac{h}{b} \approx 2.146$, which corresponds to an angle of approx. $65°$.

6. **Geometric Relation Analysis:** The model reconsidered potential geometric shortcuts, distrusted its calculation, and guessed a different relation, concluding that $\angle 2 = 20°$.

**Final Answer:** $\boxed{20°}$ **(Incorrect)**

Figure 4: Case Study: Deepseek R1 without Our Method. Despite receiving perfectly parsed visual information, the model defaults to a complex coordinate geometry approach. It makes calculation errors or loses confidence in its complex derivation, ultimately resorting to a guess and providing an incorrect answer.

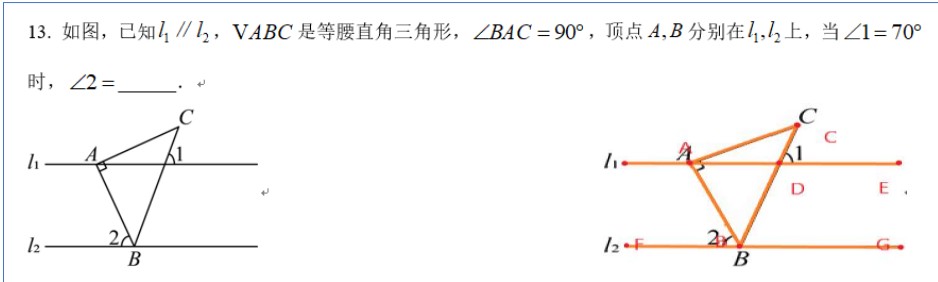

13. 如图，已知 $l_1 /\!/ l_2$，V$ABC$ 是等腰直角三角形，$\angle BAC = 90°$，顶点 $A, B$ 分别在 $l_1, l_2$ 上，当 $\angle 1 = 70°$ 时，$\angle 2 = $______.

**Geometry Problem:** As shown in the figure, given $l_1 \parallel l_2$, $\triangle ABC$ is an isosceles right-angled triangle with $\angle BAC = 90°$. Vertices $A$ and $B$ are on lines $l_1$ and $l_2$ respectively. When $\angle 1 = 70°$, find the measure of $\angle 2$.

**Image Parsing Result:** ...    (Correct structured data)   ...
**Fact Deduction Result:** $\angle 2 + \angle ABC + \angle CBG = 180°$...

**Key Steps:**

1. **Isosceles Right Triangle Properties:** Since $\triangle ABC$ is an isosceles right triangle with $\angle BAC = 90°$, we have $AB = AC$ and $\angle ABC = \angle ACB = 45°$.

2. **Parallel Lines Properties:** Let the intersection of the extension of BC with $l_1$ be D. The given $\angle 1 = 70°$ is the angle between line CD and line $l_1$. Since $l_1 \parallel l_2$, the corresponding angle at line $l_2$ is also 70°.

3. **Angle Relationships at Vertex B:** Let's denote the angle formed by the line segment BC and the rightward direction of line $l_2$ as $\angle CBG$. Due to the parallel lines, $\angle CBG = \angle 1 = 70°$.

4. **Calculation:** The angles on the straight line $l_2$ around point B must sum to $180°$. We have the angle formed by segment AB and the leftward direction of $l_2$, which is $\angle 2$. The angles are adjacent on a straight line: $\angle 2 + \angle ABC + \angle CBG = 180°$ Substituting the known values: $\angle 2 + 45° + 70° = 180°$ Solving for $\angle 2$: $\angle 2 = 180° - 45° - 70° = 65°$

**Final Answer:** $\boxed{65°}$ **(Correct)**

Figure 5: Case Study: Our Method. Our method provides supplementary guidance, prompting the model to leverage deductive geometric reasoning. This guides the model to a simple and correct solution based on angle relationships, avoiding the pitfalls of the less-interpretable coordinate method.

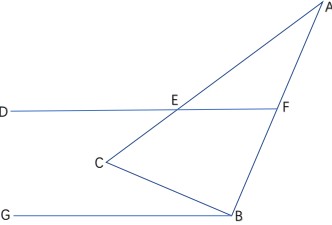

Figure 6: Example problem for applying M-theorem.

A.3 PROMPT DESIGN

---

**Geometric Relationship Prompts in Chinese and English**

**English Version:**
Please describe the geometric information in the following image, including spatial logical relationships: collinearity, concyclicity, angle relationships; pixel coordinates of the points, and geometric inference. The format is as follows:
**Analysis of the Image:**
**Spatial Logical Relationships**
- Concyclic relationship (points strictly in counter-clockwise order): [Example: Points A, B, C are concyclic, lying on a circle with O as its center.]
- Collinear relationship (points strictly in left-to-right and top-to-bottom order): [Example: Points G, F, D are collinear; Points A, G, B are collinear; Points B, E, C are collinear.]
- Perpendicular relationship: [Example: Line AGB is perpendicular to line AD; Line AGB is perpendicular to line BEC; Line DC is perpendicular to line BEC; Line AD is perpendicular to line DC.]
- Parallel relationship: [Example: Line GH is parallel to line ED; Line AD is parallel to line BEC.]
- Angle relationship: [Example: ∠C and ∠BCA are the same angle.]
**Pixel Coordinates of Points (for positional judgment only, not for numerical calculations)**
[Example of coordinates: B: x=147, y=30 O: x=90, y=77 A: x=32, y=29 C: x=122, y=146]
**Geometric Inference**
[Example of geometric inference: - The inscribed angle theorem: ∠ACB = ∠AOB / 2. - Alternate interior angles are equal: ∠CED = ∠ADE, ∠EDG = ∠DGH.]

中文版本:
请描述以下图片中几何图形的信息,包括空间逻辑关系:共线、共圆,角度关系;像素点坐标,以及图形解析推论,格式如下:
图片的解析结果:
空间逻辑关系
- 共圆关系(点严格按照逆时针顺序):[共圆关系示例:点A、B、C共圆,位于以O为圆心的圆上。]
- 共线关系(点严格按照从左到右再从上到下的顺序):[共线关系示例:G、F、D 3点共线; A、G、B 3点共线; B、E、C 3点共线。]
- 垂直关系:[垂直关系示例:直线AGB与直线AD垂直;直线AGB与直线BEC垂直;直线DC与直线BEC垂直;直线AD与直线DC垂直。]
- 平行关系:[平行关系示例:直线GH与直线ED平行;直线AD与直线BEC平行。]
- 角度关系:[角度关系示例:∠C和∠BCA是同一个角。]
点像素坐标
[点坐标示例:B: x=147, y=30 O: x=90, y=77 A: x=32, y=29 C: x=122, y=146]
图形解析推论
[图形解析推论示例: -圆周角定理:∠ACB度数=∠AOB度数的一半。 -内错角相等:∠CED=∠ADE,∠EDG=∠DGH。]

---

**Problem-Solving Prompts in Chinese and English**

**English Version:**
Please solve the following math problem. Provide a step-by-step explanation, including the problem analysis, followed by the solution steps, and finally the answer. Use the following output format: **Problem Analysis**, **Solution Steps**, **Answer**

中文版本:
请解答以下数学问题。请一步一步作答,给出题目分析,然后解题步骤,最后给出答案。输出格式如下:**题目分析**, **解题步骤**, **答案**

---

---

**Problem-Solving Prompts with Geometric Information in English**

**English Version:**
Please strictly follow the requirements below to accurately solve the geometry problem:
* Task Requirements
** Step 1: Attempt to Reconstruct the Diagram
1. If the diagram can be directly reconstructed from the "problem text" alone, proceed directly to Step 2.
2. If the "problem text" alone is insufficient to fully reconstruct the diagram, use the "spatial logical relationships" and "pixel coordinates of the points" from the "diagram information" to determine the relative spatial relationships. Note that "pixel coordinates" should only be used as a reference for reconstructing the diagram and must **not** be used for solving or calculating.
3. If there is a conflict between the diagram information and the problem description, the "problem text" takes precedence.
** Step 2: Analyze and Solve Each Sub-question
For each sub-question in the problem:
1. Clearly identify the common conditions shared by the problem statement and those specific to this sub-question.
2. Determine whether conclusions from previous sub-questions can be used, but only if the conditions match.
3. If the diagram structure changes based on the problem's conditions, re-construct the diagram based on both the "diagram information" and the problem description.
4. Use the "diagram information" to perform geometric analysis, reasoning, and detailed calculations, and provide the final answer for the sub-question.
* Problem Text
[Problem Text]
* Diagram Information
** Diagram Analysis Result:
[Geometry Problem Formalization Module Result]
** Diagram Analysis Inferences
[Geometric Theorem Reasoning Result]
* Output Requirements
For each question, provide a clear and structured problem-solving process:
1. Clearly list the conditions used. If conclusions from earlier parts are referenced, explain why.
2. If necessary, explain how key information is extracted from the diagram.
3. Show key steps and necessary calculations, providing a clear answer.
4. If there are multiple sub-questions, answer them individually and summarize all answers at the end.

972
973
974
975
976
977
978
979
980
981
982
983
984
985
986
987
988
989
990
991
992
993
994
995
996
997
998
999
1000
1001

## Problem-Solving Prompts with Geometric Information in Chinese

中文版本：
请严格遵循以下要求，准确解答几何问题：
\* 任务要求
\*\* 步骤一：尝试还原图形
1、若仅凭"题目文字"可直接还原图形，则直接跳至步骤二。
2、若"题目文字"无法完整还原图形，请结合"图形信息"中的"空间逻辑关系"和"点像素坐标"信息，来确定相对空间位置关系，注意"点像素坐标"仅用来协助还原图形，禁止用于求解计算。
3、若图像信息和题目内容存在冲突，以题目文字为准。
\*\* 步骤二：逐题分析与求解
若题目包含多子问，请对每个子问：
1、明确题干公用条件和本问特有条件;
2、判断是否可引用前面子问结论，仅在条件一致时可复用;
3、若图形结构随题设条件变化，应重新结合"图形信息"与文字进行图形重建;
4、可参考对应"图形信息"中"图形解析推论"，进行思路分析 + 详细计算，给出该子问最终答案。
\* 题目文字
[题目文字]
\*\* 图形信息
\*\* 图1解析结果:
[题图解析模块结果]
\*\* 图形解析推论
[几何定理推导结果]
\* 输出要求
针对每个问题，提供清晰结构化解题过程：
1. 明确列出所用条件，若引用前面结论，请说明;
2. 必要时说明如何从图形中提取关键信息;
3. 展示关键步骤和必要的计算过程，给出明确答案;
4. 若有多个子问，逐一解答并在最后统一总结所有答案。
