# OpenReview forum: "From Symbolic Perception to Logical Deduction: A Framework for Guiding Language Models in Geometric Reasoning"
_ICLR.cc/2026/Conference — ICLR 2026 Conference Withdrawn Submission_

### Official Review · Reviewer_ag9d · 2025-10-23

**Soundness:** 3
**Presentation:** 3
**Contribution:** 2
**Rating:** 2
**Confidence:** 4

**Summary:**

The authors focus on solving geometry problems with both images and text descriptions. They propose a pipeline that first uses a diagram parser to convert the images into structured representations, and then a symbolic solver trying to solve the problems and then LLM to generate the final answer. They also provide a new dataset of geometry problems from ZhongKao exams with images. Experiments show that their method outperforms some state-of-the-art LLMs on this dataset.

**Strengths:**

1. The authors propose a practical pipeline to convert the geometry problems with images into a format that LLMs can better understand and solve.
2. Their methods can be easily combined with any LLMs without changing the model structure and requirement to re-train the model.
3. They provide a new dataset of geometry problems from ZhongKao exams with images.

**Weaknesses:**

1. The authors claim their ZhongKao problems are more difficult than the existing geometry problems. However, there are many new datasets which are significantly more difficult than ZhongKao, such as geometry problems in math olympiad (e.g., [IMO-geometry](https://huggingface.co/datasets/theblackcat102/IMO-geometry), the geometry parts in [OMNI-math](https://huggingface.co/datasets/KbsdJames/Omni-MATH) and [GAOKAO](https://github.com/OpenLMLab/GAOKAO-Bench)). The scale of the Zhongkao is also not larger than these datasets.
2. The experiments do not show significant improvement over the existing methods. Considering the method is based on DeepSeek-R1, the improvement is quite limited (from 88.39% to 92.13%). It is also not fair to use Major@3 to compare with other methods' Major@1 results. It is commonly found that Major@3 results are significantly higher than Major@1 results for most methods including the LLMs themselves.

**Questions:**

1. Although a common believe is that iamges are important for geometric reasoning, I wonder whether it is true. In fact, there is no additional information can only be obtained from images but not from text. As my knowledge, the IMO even does not provide images for geometry problems. And the AlphaGeometry also does not require images input. So I wonder whether we need to consider the input images. Is your diagram parsing important for geometric reasoning? Can you provide some statistics about how many problems in some geometric dataset cannot deduce a unique image from only the text description? Can you show some examples to support the claim in lines 165-167: "Unlike algebraic or purely text-based tasks, geometric problem solving often depends on recognizing implicit constraints, such as collinearity, angle properties, that are visually encoded in diagrams rather than explicitly stated in text." In my experience of math olympiad, the implicit constraints do not meet the questioning standards and should be avoided very carefully.
2. It seems that the proposed method share very similar idea to the AlphaGeometry and the letter one is even introduced more interesting ideas like heuristic auxiliary lines proposed by LLMs and an iterative solving procedure. Can you compare your method with AlphaGeometry?

---

> ### Author Response · Authors · 2025-12-03
>
> We thank the reviewer for their careful reading and insightful comments. We address the concerns and questions below.
>
> ### W1
>
> Our current work focuses exclusively on the domain of plane geometry.
> The GAOKAO and IMO benchmarks also include problems involving solid (3D) geometry, which is outside the scope of our current consideration, and tackling these problems remains a key direction for our future work.
> Our test problems are definitely more difficult than those widely used plane geometry test sets such as Geometry3K, which are the focus of numerous current research efforts.
> Furthermore, many IMO problems and the GAOKAO dataset you mentioned have no images attached.
> For model capabilities, geometry problems with diagrams are more challenging than those without diagrams because they additionally involve multimodal content understanding and alignment.
>
> ### W2
>
> The metric Major@3 was included only to assess performance stability and potential gains from ensembling. All other comparative evaluations were conducted under a unified standard to ensure fairness and consistency across comparisons.
>
> ### Q1
>
> There are, in fact, many plane geometry problems where a purely textual description can lead to ambiguity.
> IMO problems typically do not have images, but many others need.
> Key information is often missing from text.
> For a simple example, a condition like "Point D is on Line AB" often leaves us unable to determine whether Point D lies inside or outside the line segment AB.
> As shown in Figure 4 of our paper, the question does not mention where angle 1 and 2 are located.
> Paper [1] also shows that geometry diagrams sharing the same textual problem, structural clauses, and semantic clauses may have different solutions.
> Moreover, works such as MATHVERSE [2] have already categorized geometry problems, including a class called "Vision Dominant" problems that inherently require integrating visual information from the diagram for correct interpretation and solving.
>
> [1] LANS: A Layout-Aware Neural Solver for Plane Geometry Problem
>
> [2] MathVerse: Does Your Multi-modal LLM Truly See the Diagrams in Visual Math Problems?
>
> ### Q2
>
> The fundamental difference lies in the dependency structure:
> In AlphaGeometry, the LLM serves as an assistant, with the primary problem-solving reliance placed on the symbolic solver.
> In our approach, the symbolic solver serves as the assistant, with the primary problem-solving reliance placed on the LLM's generative and reasoning capabilities, guided by accurate geometric perception.

---

### Official Review · Reviewer_Vqsy · 2025-10-30

**Soundness:** 2
**Presentation:** 3
**Contribution:** 2
**Rating:** 4
**Confidence:** 3

**Summary:**

This paper tackles plane geometry problem solving via  hybrid neuro-symbolic framework enabling a pure LLM (DeepSeek-R1) to perform on par with SOTA LMMs such as Gemini 2.5 Pro.

**Strengths:**

-  clear design of three-stage system for plane geometry

- new dataset that emphasizes multi-step and multi-objective questions

- ompetitive empirical results on Gemini 2.5 Pro on L2/L3 (especially with Major@3).

**Weaknesses:**

- No formal basis of the proposed Literal Expansion, Fact Deduction and Fact Filtering. What is the sysntac, semantics and detuctive reasoning solver used for this. What are the theorical guarentees? What are the guarentee od soundness and completeness of the derived facts with respect to a declared axiom?

- The fact-deduction engine relies on ordering constraints and geometric priors to prune a combinatorial search. Does the pruning preserves completeness for the targeted theorems?

- The system “translates into FormalGeo,” while Appendix tables list predicates, there is no full semantics.

- The solver chooses theorems via search + filters. It is not clear how the the matching criterion, unification and substitutions work.

-Evaluation design is somehow weak. Baselines are used with single run making Major@3 not fair for comparaison? Cost/latency should be reported.

**Questions:**

see weaknesses.

---

> ### Author Response · Authors · 2025-12-03
>
> We thank the reviewer for their careful reading and insightful comments. We address the concerns and questions below.
>
> ### W1
>
> We agree that a deeper theoretical explanation of the underlying success factors is important. However, providing a comprehensive theoretical analysis is beyond the scope of the current paper. Nevertheless, we would be very willing to discuss this issue with interested scholars in future work.
>
> ### W2
>
> Yes. Our primary pruning strategy involves filtering out options that do not meet the input requirements or preconditions of the geometric theorems, thereby significantly reducing the search space and computational overhead.
>
> ### W3, W4
>
> Thank you for your constructive suggestions. We acknowledge the value of the recommended additions. We plan to incorporate this content in a subsequent revision of the manuscript.
>
> ### W5
>
> The metric Major@3 was included only to assess performance stability and potential gains from ensembling. All other comparative evaluations were conducted under a unified standard to ensure fairness and consistency.

---

### Official Review · Reviewer_ws1b · 2025-11-01

**Soundness:** 2
**Presentation:** 2
**Contribution:** 2
**Rating:** 2
**Confidence:** 5

**Summary:**

This paper proposes a framework of to solve plane geometry problems: first, use a geometric visual parser to convert the points/lines/circles/angles in the image into structured symbol representations, and align them with the question text.
Then use a symbolic reasoning tool to match and prune key properties such as angles, generating high confidence geometric facts. Finally, feed the obtained information as an "enhanced context" to a general LLM to generate readable solutions. The author also proposed the "ZhongkaoGeo" three-layer benchmark (L1/L2/L3), emphasizing the use of exam questions in the near future to reduce training pollution, and reported comparable or better results with Gemini 2.5 and other LLMs.

**Strengths:**

1.	This paper propose a framework by firstly formalize the problem and further conduct theorem reasoning to facilitate LLMs on the problem solving.
2.	A new benchmark ZhongkaoGeo was proposed, to challeng that current benchmarks facing the data leaking issue to the training of LLMs, and also has different layers to classify the difficulty of the problems.

**Weaknesses:**

1.	**Limited novelty and contribution.** The framework mainly integrates existing components (YOLO detection, rule-based symbolic solver from FormalGeo, and prompt-based LLM reasoning) without introducing a new learning paradigm or optimization objective. And the theorem reasoning module leveraged FormalGeo. It remains unclear how these modules interact under a unified theoretical framework or whether the system optimizes any well-defined objective. As a result, the contribution appears more engineering-driven than learning-oriented, which weakens its theoretical novelty for an ICLR submission.
2.	The authors claim that providing verified symbolic facts can “effectively constrain the reasoning space and mitigate hallucination.” (Line 283, 284) However, the paper lacks a theoretical mechanism explaining this claim. Author does not define what the “reasoning space” is, nor describe how it is constrained by symbolic facts, and provides no formal framework (e.g., probabilistic, information-theoretic, or search-based) to support the notion of such space reduction.
3.	**Limited parsing ability and query of the propose dataset.** The current parser seems has no ability to understand the annotations in the diagram, like the degree of the angle, length of line. It is crucial for solving PGPs, if the problems in ZhongkaoGeo do not require the annotation understanding, that means the dataset is too simple, which not satisfy the current research requirements on PGP area. The previous work like PGDP [1] already tackled the parsing task with very high accuracy, the parsing module in this work has no contribution to the study of plane geometry diagram parsing.
4.	The experiment is not fair. The author used major@3 to get performance increase, under this case, other baselines should also use the majority vote in the experiments. Comparing with the performance without majority vote, the proposed method has lower performance then Gemini2.5-Pro in ZhongKaoGeo-L1.
5.	**Experiments are not sufficient.** It is necessary to conduct experiments on popular benchmarks such as Math-Verse, Math-Vista, the current experiments only conduct on the proposed dataset with very limited LLMs. It is not related to the data leaking issue, as the framework is universal to different models, I just wondering the effectiveness of the framework. Also, the proposed method is a framework, it is necessary to apply this framework with different LLM backbones, not only limited to DeepSeek-R1, like assemble your framework with Gemini, GPT, to show whether the performance will increase based on this framework.
6.	The problems in ZhongkaoGeo has several types (angle, length, area …). As a benchmark, these statistics of the experiments should be introduced, the current table is too simple, only has accuracy. No deeper analyses were provided from the experiments.
7.	Despite the author claim “Large Multimodal Models (LMMs) such as Gemini 2.5 Pro handles visuo-linguistic inputs but are resource-intensive.”, experiments should also cover the experiments of LMMs with smaller parameters, like many plane geometry problem solving models, such as R-CoT, G-LLaVA, GeoX. Comparing to models like Gemini, these models use lower computing resources. What is the performance of these models on ZhongkaoGeo? If they has better performance, the value of this framework will be challenged.
8.	**Losing details.** How did you train the YOLO to do the parsing task, what dataset used to train this module?
Above all, the current version of this paper is not satisfy the bar of ICLR.

Above all, the current version of this paper is not satisfy the bar of ICLR.

Ref:

[1] Plane Geometry Diagram Parsing, in IJCAI 2022

**Questions:**

1．	Line 049, “This paper challenges the prevailing assumption that end-to-end LMMs are the definitive solution for complex geometric reasoning. “ Is there any source to support this claim? As far as I know, now a popular research direction in solving PGPs is not end-to-end, like a lot of works are using neural-symbolic framework.

2．	There are L1, L2 and L3 in ZhongkaoGeo, why the statistic of L3 not be introduced in Table 1?

---

> ### Author Response · Authors · 2025-12-03
>
> We thank the reviewer for their careful reading and insightful comments. We address the concerns and questions below.
>
> ### W1
>
> We respectfully disagree with the assessment that our work is a simple integration. In fact, we have meticulously designed the components.
> We did not merely use the raw output of a YOLO model. Our design involved a series of steps including post-training, pipeline design, and specific post-processing to ensure accurate and tailored geometric perception.
> Similarly, the symbolic reasoning module was carefully engineered to integrate closely with our overall framework, rather than being a rigid, standalone component, allowing it to seamlessly contribute to the hybrid solution process.
>
> ### W2
>
> We prevent future reasoning sequences that are wrong, by providing better context to the LLM. This context (given in prompts) are the relationships extracted from the image using our CV modules, higher order relationships and other deduced facts. because of this we lower the chances the model may hallucinate these facts and thus prevent "bad" reasoning sequences from occuring.
>
> We agree that a deeper theoretical explanation of the underlying success factors is valuable. However, providing a comprehensive theoretical analysis is beyond the scope of the current paper, which focuses on a practical system implementation and empirical results. Nevertheless, we would be very willing to discuss this issue with interested scholars in future work.
>
> ### W3
>
> Our parser is fully capable of understanding the annotations in the diagram. We are puzzled by this misunderstanding and will take immediate steps to revise the description and presentation of the parser in the manuscript to prevent future confusion.
> Regarding PGDP, while it has shown strong results on datasets like Geometry3K and PGDP5K, its performance falls short on more complex plane geometry problems, particularly those found in the ZhongKao benchmark. This disparity is precisely the motivation for re-proposing and designing our specialized parsing module to handle complex, real-world geometric figures robustly.
>
> ### W4
>
> All comparative analyses are conducted based on the same established evaluation standards. The metric Major@3 was included solely to assess performance stability and potential gains from ensembling, and not as the primary comparison metric.
>
> ### W5, W6, W7, W8
>
> Thank you for your valuable suggestions. We agree that these points would strengthen the paper. We plan to incorporate this content in a future revision of the manuscript.
>
> ### Q1
>
> The sentence in question was specifically aimed at the discussion contrasting LLMs and LMMs in the context of geometric reasoning. In the paper we immediately followed up with our core argument:
> "We posit that a LLM, when augmented with specialized geometric perception and deductive reasoning capabilities, can not only match but surpass the performance of leading LMM".
>
> Although numerous studies have applied neural-symbolic methods to solve plane geometry problems, using LMMs for this purpose has indeed become the major trend, with an increasing number of related research works emerging[1][2][3][4][5].
>
> Our claim does not imply that all prior research adopts an end-to-end paradigm. Rather, we refer to a widely held expectation in the LMM community that end-to-end multimodal models are generally capable of complex visual–geometric reasoning.
>
> [1] GeoUni: A Unified Model for Generating Geometry Diagrams, Problems and Problem Solutions
>
> [2] G-LLaVA: Solving Geometric Problem with Multi-Modal Large Language Model
>
> [3] GeoX: Geometric Problem Solving Through Unified Formalized Vision-Language Pre-Training
>
> [4] VisuoThink: Empowering LVLM Reasoning with Multimodal Tree Search
>
> [5] Geo-LLaVA: A Large Multi-Modal Model for Solving Geometry Math Problems with Meta In-Context Learning
>
> ### Q2
>
> To truly reflect the capability of various models on challenging, real-world problems, the L3 benchmark was constructed differently from L1 and L2. L3 was compiled directly from ALL plane geometry problems in the 2025 ZhongKao math exams from key provinces/cities. Due to its all-inclusive and unbiased nature of collection, we did not initially conduct a separate targeted analysis for L3. We appreciate your suggestion, nonetheless, and plan to supplement this section with a more detailed analysis of the L3 benchmark in subsequent work.

---

### Official Review · Reviewer_i5qT · 2025-11-10

**Soundness:** 3
**Presentation:** 2
**Contribution:** 3
**Rating:** 4
**Confidence:** 3

**Summary:**

This work proposes a hybrid framework to enhance the geometric reasoning ability for LLMs.
Essentially, the framework first identifies geometric elements in a given image by combining detection models and heuristic preprocessing.
Then, the framework uses a rule-based deduction engine to infer some conclusions.
Since these conclusions may not lead to the final answer of a given query, LLM is used to reason over the given problem (image + query) and the conclusions inferred by the deduction engine.

To conduct experiments, authors propose a new benchmark according to the 2025 Chinese Zhongkao examinations, ensuring complexity and avoiding data contamination.
On the proposed benchmark, results show that the proposed framework can achieve performance comparable to large multimodal models.

**Strengths:**

1. The paper is well-written and well-organized.
2. The proposed framework and benchmark are well-motivated.

**Weaknesses:**

1. Experiments are not enough. Authors should compare the proposed framework with other symbolic or hybrid methods. Although authors claim that involving symbolic systems has many issues (Line 35-41), the proposed framework also involves symbolic conversion, symbolic deduction, and heuristic processing. Therefore, the proposed framework also suffers from many of those issues and should be compared with other symbolic or hybrid methods.
2. Figure 3 is not good. For example, the left subfigure is too big with just a little information. The fonts in the subfigures also looks non-academic.
3. The presentation is not clear enough. See *Questions*.

**Questions:**

1. Line 45-48: You claim that LMMs are computationally expensive, resource-intensive, and their reasoning can be opaque, and that their performance on specialized domains like geometry is often constrained by distributional shifts between their generalist training data and the specific symbolic logic of geometric diagrams. Are not these also limitations of LLMs?
2. Line 46: What is resource-intensive? What is the difference between resource-intensive and computationally expensive?
3. Line 76-77: Is there evidence justifying that the deduction on angular relationships is a frequent source of LLM hallucination? The description given in Line 243-249 is not convincing. Real examples that can justify the description in Line 243-249 will be more convincing.
4. Table 5: Is the degree of angle annotated in the original image from the dataset? Is so, recognizing the degree from the images can show that Gemini-2.5-Pro has strong image understanding ability, instead of data contamination.
5. Line 172-173: Why LLMs can produce more coherent solutions than symbolic solver?
6. Is the proposed framework only capable of addressing angular relation problem?
7. How does the difficulty of the benchmark you proposed compare to existing benchmarks?

---

> ### Author Response · Authors · 2025-12-03
>
> We thank the reviewer for their careful reading and insightful comments. We address the concerns and questions below.
>
> ### W1
>
> We acknowledge that we pointed out some limitations of purely symbolic methods in solving plane geometry problems.
> For example, these other symbolic solvers are standalone and do not rely on LLMs to further refine their results.
> However, this does not imply that symbolic methods are useless.
> In fact, the exact reasoning provided by symbolic methods is one of the key elements for the success of our solving framework.
> On the other hand, our approach uses LLMs as the second main step in the pipeline helping with translated the results from the symbolic engine to natural human-readable language.
> Our core objective is to integrate the strengths of both symbolic methods and LLMs to achieve superior performance in plane geometry problem-solving, thereby enabling us to challenge the more powerful LMMs.
>
> ### Q1
>
> Yes, we agree that this is a known issue for LLMs, which is precisely why we ultimately selected a hybrid approach that leverages both symbolic and generative models.
>
> ### Q2
>
> In the context of this paper, the meanings of these two terms are consistent.
>
> ### Q3
>
> This conclusion is drawn from our extensive analysis of a large number of problems. We plan to further organize these results and present this evidence in a future version of the manuscript. Furthermore, other works, such as [1], have provided similar findings.
>
> [1] DO LARGE LANGUAGE MODELS TRULY UNDERSTAND GEOMETRIC STRUCTURES?
>
> ### Q4
>
> It appears the reviewer may have overlooked the Caption of Table 5. The problem image was not provided to the Gemini model for this specific evaluation.
>
> ### Q5
>
> We originally intended to express that the generative capability of the LLMs allows it to explain ‘why a specific step of reasoning is taken’ outside of the rigid logical inference chain. This results in solutions that are easy to understand and highly valuable for teaching or self-study.
> In the revised version, we will remove the term “coherent” to prevent any potential misunderstanding.
>
> ### Q6
>
> No, this is incorrect. Our framework is designed to solve a variety of plane geometry problems, not just those focused on finding angles. It handles problems involving lengths, areas, and various geometric properties.
>
> ### Q7
>
> Our proposed benchmark is superior not only because it demands a greater number of reasoning steps (making it inherently more difficult), but also because its derivation from the latest exams ensures a significantly lower risk of data leakage compared to older or commonly used datasets.

---

### Note · Authors · 2025-12-04

I have read and agree with the venue's withdrawal policy on behalf of myself and my co-authors.